# COUNTDOWN REGRESSION:
# SHARP AND CALIBRATED SURVIVAL PREDICTIONS

## ABSTRACT

Personalized probabilistic forecasts of time to event (such as mortality) can be crucial in decision making, especially in the clinical setting. Inspired by ideas from the meteorology literature, we approach this problem through the paradigm of maximizing sharpness of prediction distributions, subject to calibration. In regression problems, it has been shown that optimizing the continuous ranked probability score (CRPS) instead of maximum likelihood leads to sharper prediction distributions while maintaining calibration. We introduce the *Survival-CRPS*, a generalization of the CRPS to the time to event setting, and present right-censored and interval-censored variants. To holistically evaluate the quality of predicted distributions over time to event, we present the scale agnostic *Survival-AUPRC* evaluation metric, an analog to area under the precision-recall curve. We apply these ideas by building a recurrent neural network for mortality prediction, using an Electronic Health Record dataset covering millions of patients. We demonstrate significant benefits in models trained by the Survival-CRPS objective instead of maximum likelihood.

## 1 INTRODUCTION

Having patient-specific predictions of time to an event such as mortality or bone fracture allows caregivers to make better informed decisions around patient care. Historically, prognosis scores have served as simple tools to stratify patient risk within a predefined time window (Lau et al., 2006; Cardona-Morrell & Hillman, 2015). However, such models tend to be too simplistic to be widely useful. They are often estimated from a large population of patients, and do not take into account patient-specific information to make individualized predictions (Yu et al., 2011). Meanwhile, the adoption of Electronic Health Record (EHR) systems over the past few decades has resulted in the collection of observational data on millions of patients spanning multiple years. This data enables development of patient-specific prediction models using machine learning. Such models are applicable to the larger patient population without being specific to a disease type or demographic, and this makes it possible to develop novel workflows in care delivery. For example, a high predicted probability of 3-12 month mortality could proactively notify palliative care teams of otherwise overlooked patients with end-of-life needs (Avati et al., 2017).

One way to obtain patient-specific survival predictions is to treat the problem as probabilistic classification; that is, training a binary classifier to predict outcomes of event by a particular time of interest (Avati et al., 2017; Rajkomar et al., 2018). However, such an approach has drawbacks. First, the model is specific to the time of interest it was trained upon – it is not straightforward how to take a model that was trained to predict probabilities of 1-year mortality and obtain predictions of 6-month mortality from it. Second, it is not usually possible to use data on all patients – for example, if a patient has only 3 months of history in the EHR system, it is neither possible to include that patient as a positive case nor a negative case in the 1-year mortality prediction task. Third, the process of constructing the data set implicitly conditions on the future outcome to select prediction times – evaluation is performed only at times looking backward from the event of interest. It has been shown that evaluation metrics can be overly optimistic relative to real world performance as a result (Sherman et al., 2017).

An alternative approach to the problem is survival prediction; that is, predicting time to event by estimating a distribution over future time. In this setting, traditional survival analysis methods such

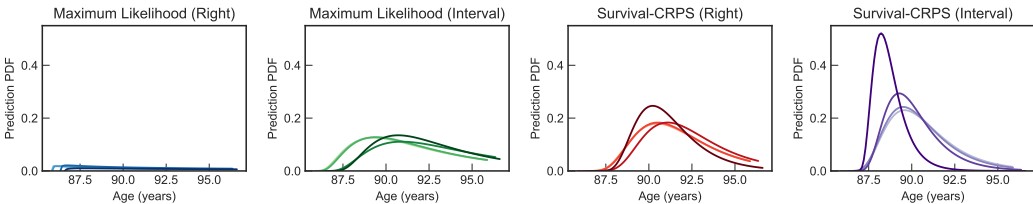

Figure 1: Example of a patient's predicted distributions for age of death under different models. Our proposed techniques improve sharpness of predicted distributions, subject to calibration. Repeated interactions (indicated by darker color) between the patient and the EHR yield more confident predictions of time of death.

as the Cox proportional hazards model (R., 1972) or accelerated failure time models (J.) are capable of handling data with censored observations (cases in which the event was not observed, but we know that the event did *not* occur up to a certain time). This addresses concerns raised by the classification approach, but there are a few nuances. First, traditional models typically make strong assumptions, such as proportional hazards or linearity. Second, challenges of low prevalence often arise when these methods are applied to large-scale observational datasets with heavy censoring, which is the case in real EHR data. Third, these survival analysis methods are typically evaluated as point estimates of risk, such as 10-year probabilities of events, rather than holistic measures of quality of the predicted distributions (Goff et al., 2014; Ranganath et al., 2016; Lee et al., 2018). Common metrics of evaluation include the C-statistic (Uno et al., 2007), log-$\ell_1$ loss (Yu et al., 2011), and mean-squared-error (Katzman et al., 2018). While useful for the purposes of relative risk stratification, model comparisons made using point estimates leaves the quality of uncertainty in predicted distributions left unmeasured. If a point prediction is way off, it is penalized by the same amount whether the model was confident or not (that is, whether the predicted distribution had low or high variance).

In contrast, forecasts in the field of meteorology are typically made as full prediction distributions over all weather conditions given past and current observations (Gneiting et al., 2008). Evaluation of predictive performance is assessed by the paradigm of maximizing the sharpness of the predictive distribution, subject to calibration (Gneiting & Katzfuss, 2014). The intuition behind this paradigm is that probabilities have to be *calibrated* in order to be correct. However, that does not necessarily make them useful (one could always predict the marginal probability of an outcome without looking at the data, and still be well calibrated). The usefulness of a prediction distribution lies in its *sharpness*, or how well its mass concentrates. In summary, uncalibrated predictions (sharp or not) are useless, calibrated but non-sharp predictions are correct but less useful, and calibrated and sharp distributions are most useful.

To improve the sharpness of prediction distributions in the survival setting, we propose the use of proper scoring rules beyond maximum likelihood as the training objective. Proper scoring rules are known to measure calibration, and any model trained with a proper scoring rule will tend to maintain calibration (Gneiting & Katzfuss, 2014). For our purposes, we focus on the *continuous ranked probablity score* (CRPS) which has been used as an objective in the regression setting (Gneiting et al., 2008; Mohammadi et al., 2016; 2015). We generalize the CRPS for the survival setting, called Survival-CRPS, with right-censored and interval-censored extensions. Our work is one among many recent works Chapfuwa et al. (2018) in using non-MLE training objectives for survival prediction.

*Summary of contributions.* (1) We introduce the proper scoring rule Survival-CRPS, a generalization of CRPS, as an objective in survival prediction. We present its right-censored and interval-censored variants. (2) We propose a new metric, Survival-AUPRC, inspired by the paradigm of maximizing sharpness subject to calibration, to holistically measure the quality of a prediction distribution with respect to a possibly censored outcome. (3) We give practical recommendations for the mortality prediction task, by recommending use of the log-normal parameterization and interval censoring when training. (4) We employ the above techniques and demonstrate their efficacy by training a deep recurrent neural network model for accurate survival prediction of patient mortality using EHR data.

## 2 COUNTDOWN REGRESSION

Parametric survival prediction methods model the time to an event of interest with a family of probability distributions, indexed by the distribution parameters. The survival function, denoted $S(t) : [0, \infty) \to [0, 1]$, is a monotonically decreasing function over the positive reals with $S(0) = 1$ and $\lim_{t \to \infty} S(t) = 0$. The survival function represents the probability of an individual not having the event of interest up to a given time. Every survival function has a corresponding cumulative density function (CDF), denoted $F(t) = 1 - S(t)$, and probability density function (PDF), denoted $f(t) = \frac{d}{dt} F(t)$. The choice of the family of probability distributions implies assumptions made about the nature of the data generating process.

We denote the medical record of a patient $i$ as $\left( \{(x_t^{(i)}, a_t^{(i)})\}_{t=1}^{T^{(i)}}, d^{(i)}, c^{(i)} \right)$, where $t \in \{1 \ldots T^{(i)}\}$ denotes the interaction number of this patient with the health record, $x_t^{(i)} \in \mathbb{R}^D$ is the set of features corresponding to the $t$-th interaction, $a_t^{(i)} \in \mathbb{R}_+$ is age at time $t$, $d^{(i)} \in \mathbb{R}_+$ is the age of death or age of last known (alive) encounter, and $c^{(i)} \in \{0, 1\}$ is a censoring indicator where $c^{(i)} = 0$ means the age of death is $d^{(i)}$, and $c^{(i)} = 1$ means the age of death is at least $d^{(i)}$. For each $x_t^{(i)}$ we define the quantity $y_t^{(i)} = d^{(i)} - a_t^{(i)}$ which represents the corresponding time to event or time to censoring.

Traditional methods in survival analysis are designed to handle right-censored outcomes, but we observe that in many common scenarios outcomes are actually interval-censored. In the context of mortality prediction, for example, we know that humans almost never live past 120 years of age. Therefore, we assume that the true age of death lies between $d^{(i)}$ and $\mathcal{A} = 120$ years, implying that the true time to death lies between 0 and $\mathcal{T}_t^{(i)} = \mathcal{A} - a_t^{(i)}$. We omit patient superscripts $i$ and interaction subscripts $t$ for succinctness where possible. We note that although our notation focuses on the problem of mortality prediction, our techniques generalize to any time to event task of interest.

### 2.1 SURVIVAL-CRPS: PROPER SCORING RULE OBJECTIVES

A scoring rule is a measure of the quality of a probabilistic forecast. A forecast over a continuous outcome is a probability density function over all possible outcomes, $\hat{f}$ with corresponding cumulative density function $\hat{F}$. In reality, we observe some actual outcome, $y$. A scoring rule $\mathcal{S}$ takes a predicted distribution and an actual outcome, and returns a loss $\mathcal{S}(\hat{F}, y)$. It is considered a *proper scoring rule* if for all possible distributions $G$,

$$\mathbb{E}_{y \sim \hat{F}}[\mathcal{S}(\hat{F}, y)] \leq \mathbb{E}_{y \sim \hat{F}}[\mathcal{S}(G, y)],$$

and *strictly* proper when equality holds if and only if $\hat{F} = G$ (Gneiting et al., 2008). A proper scoring rule is one in which the expected score is minimized by the distribution with respect to which the expectation is taken. Intuitively, it encourages a model for being honest by predicting what it actually believes (Savage, 1971). When a proper scoring rule is employed as a loss function, it naturally forces the model to output calibrated probabilities (Gneiting & Katzfuss, 2014).

There are many commonly used proper scoring rules. Perhaps the most widely used is the logarithmic scoring rule, equivalent to the maximum likelihood objective:

$$\mathcal{S}_{\text{MLE}}(\hat{F}, y) = -\log \hat{f}(y).$$

In the presence of possibly censored data, we maximize the density for observed outcomes, and tail or interval mass for censored outcomes, and this is a proper scoring rule (Dawid & Musio, 2014).

$$\mathcal{S}_{\text{MLE-RIGHT}}(\hat{F}, (y, c)) = -\log \left( (1 - c)\hat{f}(y) + c\hat{S}(y) \right)$$

$$\mathcal{S}_{\text{MLE-INTVL}}(\hat{F}, (y, c, \mathcal{T})) = -\log \left( (1 - c)\hat{f}(y) + c(\hat{F}(\mathcal{T}) - \hat{F}(y)) \right)$$

However, the logarithmic scoring rule is asymmetric, and harshly penalizes predictions that are wrong yet confident. This results in the training process becoming sensitive to outliers, and in general conservative in prediction-making (that is, hesitant to make sharp predictions) (Gneiting & Raftery, 2007).

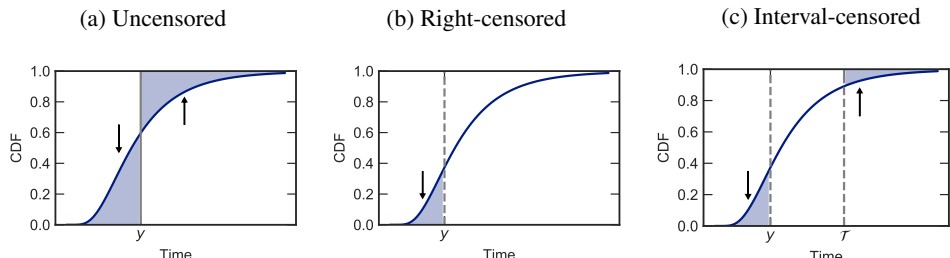

Figure 2: Graphical intuition for the Survival-CRPS scoring rule. For uncensored observations, we minimize mass before and after the observed time of event. For right-censored observations, we minimize mass before observed time of censoring. For interval-censored observations, we minimize mass before observed time of censoring, and mass after the time by which event must have occurred.

Another proper scoring rule for forecasts over continuous outcomes is the CRPS (Gneiting et al., 2007), defined as

$$\mathcal{S}_{\text{CRPS}}(\hat{F}, y) = \int_{-\infty}^{\infty} \left( \hat{F}(z) - \mathbb{1}\{z \geq y\} \right)^2 dz = \int_{-\infty}^{y} \hat{F}(z)^2 dz + \int_{y}^{\infty} (1 - \hat{F}(z))^2 dz.$$

The CRPS has been used in regression as an objective function that yields sharper predicted distributions compared to maximum likelihood, while maintaining calibration (Gneiting et al., 2008). Intuition for the CRPS is better understood by analyzing the latter expression and noting that the two integral terms correspond to the two shaded regions in Figure 2a. The CRPS score is completely reduced to zero when the predicted distribution places all the mass on the point of true outcome, or equivalently, when the shaded region completely vanishes.

In the context of time to event predictions we propose the *Survival-CRPS* which accounts for the possibility of right-censored or interval-censored data:

$$\mathcal{S}_{\text{CRPS-RIGHT}}(\hat{F}, (y, c)) = \int_{0}^{y} \hat{F}(z)^2 dz + (1 - c) \int_{y}^{\infty} (1 - \hat{F}(z))^2 dz,$$

$$\mathcal{S}_{\text{CRPS-INTVL}}(\hat{F}, (y, c, \mathcal{T})) = \int_{0}^{y} \hat{F}(z)^2 dz + (1 - c) \int_{y}^{\mathcal{T}} (1 - \hat{F}(z))^2 dz + \int_{\mathcal{T}}^{\infty} (1 - \hat{F}(z))^2 dz.$$

Note that when $c = 0$, both of the above expressions are equivalent to the original CRPS. Again, the intuition behind the Survival-CRPS is better understood by mapping each of the integral terms to the corresponding shaded region in Figure 2b and Figure 2c. The Survival-CRPS behaves like the original CRPS when the time of event is uncensored. For censored outcomes, it penalizes the predicted mass that occurs before the time of censoring and, if interval censored, also the mass after time by which the event must have occurred.

Both variants of the Survival-CRPS are proper scoring rules. They are special cases of the threshold weighted CRPS (Gneiting & Ranjan, 2011), where the weighting function is an indicator over the uncensored regions.

## 2.2 EVALUATION BY SHARPNESS SUBJECT TO CALIBRATION

*Calibration* assesses how well forecasted event probabilities match up to observed event probabilities. It is crucial in development of useful predictive models, especially for clinical decision-making. In binary prediction tasks without censoring, the Hosmer-Lemeshow test statistic (Hosmer et al., 2011) is commonly used to assess goodness-of-fit by comparing observed versus predicted event probabilities at quantiles of predicted probabilities. Extensions to account for censoring have been proposed (Grønnesby & Borgan, 1996; D'Agostino & Nam, 2003; Demler et al., 2015), but these methods apply only to predictions of dichotomous outcomes within a particular time frame (for example, 1-year risks of mortality).

There is no widely accepted method for evaluating the calibration of a set of entire prediction distributions, over multiple time frames, in the survival setting. D-calibration has been recently

proposed as a method for holistic evaluation (Andres et al., 2018), but relies on handling censored observations by assuming the true times to death are uniformly distributed past the times of censoring in the predicted distributions. When censored observations far outnumber the uncensored observations, this can lead to overly optimistic assessments of calibration. Another option is to evaluate observed event times on the cumulative density scale of predicted distributions, using a Kaplan-Meier estimate to account for censoring (Harrell, 2006). Again, this method has limitations in the heavily censored setting, as the quantiles in the tail of predicted cumulative densities have few uncensored observations, and will rarely yield well calibrated values.

We instead employ the following method to measure calibration. We compare predicted cumulative densities against observed event frequencies, evaluated at quantiles of predicted cumulative density. Right-censored observations are removed from consideration in quantiles that correspond to times after their points of censoring. Interval-censored observations are similarly removed from consideration in quantiles that correspond to times after censoring, but are additionally re-introduced in quantiles that correspond to times past the time by which the event must have occurred (in the mortality prediction task, this corresponds to 120 years of age).

Subject to calibration, we strive for prediction distributions that are *sharp* (i.e, concentrated). There are several metrics that could be used for measuring sharpness, such as variance or entropy. In the context of time to event predictions, holding two distributions with vastly different means to the same standard of variance or entropy would be unfair (for example, we would want lower variance for a prediction distribution with a mean of a day, compared to a mean of a year). Instead, we use the coefficient of variation (CoV) as a reasonable measure of sharpness. The CoV is defined as the ratio of one standard deviation to the mean, $\text{CoV}(\hat{F}) = \frac{\sqrt{\text{Var}[\hat{F}]}}{\mathbb{E}[\hat{F}]}$.

### 2.3 SURVIVAL-AUPRC: HOLISTIC EVALUATION OF TIME TO EVENT PREDICTIONS

Since sharpness is only a function of the predicted distributions, a measure of sharpness is only meaningful if the model is sufficiently calibrated. We now propose a metric that measures how concentrated the mass of the prediction distribution is around the true outcome, robust to miscalibration. The idea is similar to the area under a precision-recall curve, except here it is with respect to only one predicted distribution and one outcome. We first consider the uncensored case. As an analog to precision, we consider intervals relative to the true time of event, defined by ratios. For example, a region of precision 0.9 around an event that occurs at time $y$ is the interval $[0.9y, y/0.9]$. Corresponding to this region of precision, the analogy to recall is the mass assigned by the predicted distribution over this interval, $\hat{F}(y/0.9) - \hat{F}(0.9y)$. By exploring the full range of precision from 0 to 1, we obtain the *Survival Precision Recall Curve*. The area under this curve measures how quickly predicted mass concentrates around the true outcome as we expand the precision window.

$$\text{Survival-AUPRC}_{\text{UNCENS}}(\hat{F}, y) = \int_0^1 (\hat{F}(y/t) - \hat{F}(yt))dt$$

The highest possible score is 1, when the predicted distribution is a Dirac $\delta$ function centered over the time of outcome. The lowest possible score is 0, when the predicted distribution is infinitely dispersed. The mean of all Survival-AUPRC scores across examples provides an overall measure of the quality of the predictions.

The aforementioned metric only applies when the event outcome is uncensored. In the case of censored observations, we use the same analogy but with the right end of precision intervals defined with respect to the time by which the event must have occurred in the interval-censored case, or infinity in the right-censored case.

$$\text{Survival-AUPRC}_{\text{RIGHT}}(\hat{F}, y) = \int_0^1 (1 - \hat{F}(yt))dt$$

$$\text{Survival-AUPRC}_{\text{INTVL}}(\hat{F}, y, \mathcal{T}) = \int_0^1 (\hat{F}(\mathcal{T}/t) - \hat{F}(yt))dt$$

### 2.4 RECURRENT NEURAL NETWORK MODEL

We apply our techniques to the mortality prediction task by building a multilayer recurrent neural network (RNN) with parameters $\theta$, denoted $\text{RNN}_\theta$, that takes as input a sequence of features (in our case, information about a patient recorded in the EHR, for each interaction they had with the hospital) to predict parameters of a parametric probability distribution $\hat{F}$ over time to death at each timestep (Figure 3). The network depends only on data from the current and previous timesteps, and not the future. The approach here is similar to the recently proposed Weibull time to event RNN (Martinsson, 2016), though we generalize to any choice of noise distribution. The distributions that are output in each timestep are used to construct an overall loss,

$$\mathcal{L}_{\text{RIGHT}} = \sum_{i=1}^{N} \sum_{t=1}^{T^{(i)}} \mathcal{S}_{\text{RIGHT}}\left( \hat{F}_{\text{RNN}_\theta \left\{ x_{1:t}^{(i)} \right\}}, \left( y_t^{(i)}, c^{(i)} \right) \right)$$

$$\mathcal{L}_{\text{INTVL}} = \sum_{i=1}^{N} \sum_{t=1}^{T^{(i)}} \mathcal{S}_{\text{INTVL}}\left( \hat{F}_{\text{RNN}_\theta \left\{ x_{1:t}^{(i)} \right\}}, \left( y_t^{(i)}, c^{(i)}, \mathcal{T}_t^{(i)} \right) \right),$$

where $N$ is the total number of patients in the training set, $T^{(i)}$ is the sequence length for patient $i$, and $\hat{F}_{\text{RNN}_\theta}$ denotes the distribution parameterized by the output of the RNN. It is the sequential and monotonically decreasing predicted times to event that inspires the name *Countdown Regression*.

### 2.5 CHOICE OF LOG-NORMAL NOISE DISTRIBUTION

Common parametric distributions over time to event used in traditional survival analysis models include the Weibull, log-normal, log-logistic, and gamma (in order to be sufficiently expressive in model space, we seek distributions with at least two parameters). We choose the log-normal distribution because other distributions either involve the Gamma function in their density, or involve the pattern $(y/p_1)^{p_2}$, where $p_1$ and $p_2$ are parameters output from the neural network. We found these patterns to be highly sensitive to the inputs and to suffer from numerical instability issues.

For the log-normal distribution, a closed form expression for the CRPS is well known (Baran & Lerch, 2015). However, a closed form expression for the Survival-CRPS does not exist. We perform a change of variable to express the integral terms as finite integrals, and numerically approximate with the trapezoid rule. When training, we then back-propagate through the trapezoidal approximation. Details are given in Appendix B and C. We note that the approximation formulas are themselves proper scoring rules, as they are just weighted sums of Brier scores. Closed form expressions for the log-normal Survival-AUPRC are also given in Appendix D, E, and F.

## 3 EXPERIMENTS

We run experiments for the mortality prediction task to evaluate four different training objectives: maximum likelihood $\mathcal{S}_{\text{MLE-RIGHT}}$ and $\mathcal{S}_{\text{MLE-INTVL}}$, and our scoring based loss $\mathcal{S}_{\text{CRPS-RIGHT}}$ and $\mathcal{S}_{\text{CRPS-INTVL}}$. For interval censoring we assume a maximum lifespan of $\mathcal{A} = 120$ years.

|    (a) Dead (uncens) patients.    |    (b) Alive (right-cens) patients.    |    (c) Alive (interval-cens) patients.    |

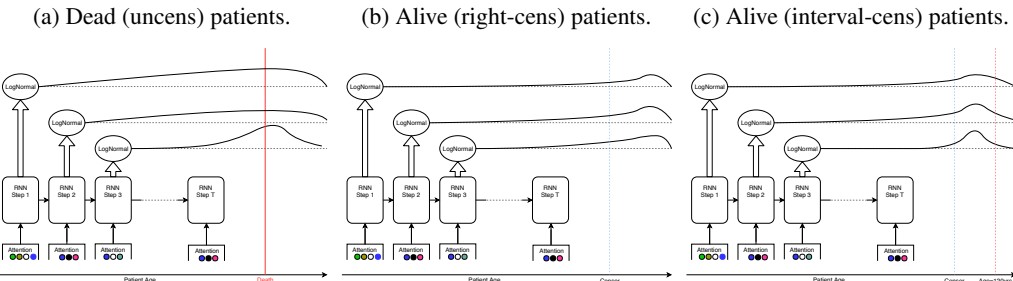

Figure 3: RNN model overview. For each interaction, we attend over recorded ICD codes at that timestep and predict parameters $\mu, \sigma^2$ of a log-normal distribution, minimizing a proper scoring rule.

Table 1: Metrics measuring sharpness and calibration for models trained on the right-censored and interval-censored variants of the maximum likelihood and Survival-CRPS objectives.

| Metric | MLE-RIGHT | MLE-INTVL | CRPS-RIGHT | CRPS-INTVL |
|---|---|---|---|---|
| Calibration slope | 1.125 ± 3e-4 | 1.139 ± 3e-4 | 1.003 ± 3e-4 | 0.959 ± 5e-4 |
| Mean coefficient of variation | 18.42 ± 5e-3 | 0.911 ± 4e-4 | 0.332 ± 1e-4 | **0.301 ± 1e-4** |
| Mean prob of survival to age 120 yrs | 0.754 ± 2e-5 | 0.045 ± 3e-5 | 0.015 ± 3e-5 | **0.005 ± 1e-6** |
| Dead: mean Surv-AUPRC (uncen) | 0.233 ± 2e-4 | 0.319 ± 3e-4 | 0.343 ± 4e-4 | **0.366 ± 4e-4** |
| Alive: mean Surv-AUPRC (intvl-cen) | 0.407 ± 6e-5 | 0.963 ± 2e-5 | **0.977 ± 3e-5** | 0.976 ± 3e-5 |

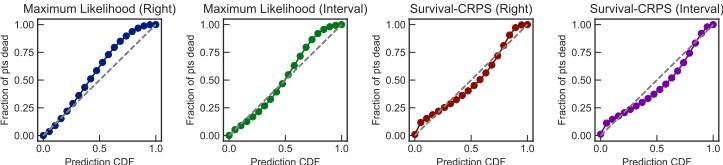

Figure 4: Calibration plots for each of the models. We compare predicted cumulative densities against observed event frequencies, evaluated at quantiles of predicted cumulative density. Right-censored observations are removed from consideration in quantiles past times of censoring, interval-censored observations are additionally re-introduced in quantiles corresponding to times past 120 years.

The neural network architecture is kept identical for all four experiments and implemented in PyTorch (Paszke et al., 2017). The input at each timestep consists of both real valued (for example, age of patient) and discrete valued (for example, ICD codes) data. Discrete data is embedded into a trainable real-valued 126-dimensional vector space, and vectors corresponding to the codes recorded at a given timestep are combined into a weighted mean by a soft self-attention mechanism. All real valued inputs are appended to the averaged embedding vector. We also provide the real valued features to every layer by appending them to the output of previous layer. The input vector feeds into a fully connected layer, followed by multiple recurrent layers. We use the Swish activation function (Ramachandran et al., 2017) and layer normalization (Ba et al., 2016) at every layer. Recurrent layers are defined using GRU units (Chung et al., 2014) with layer normalization inside. After the set of recurrent layers, the network has multiple branches, one per parameter of the survival distribution (for the lognormal, $\mu$ and $\sigma^2$). The final layer in each branch has scalar output, optionally enforced positive with the softplus function, Softplus$(z) = \log(1 + \exp(z))$. We use Bernoulli dropout (Srivastava et al., 2014) at all fully connected layers, and Variational RNN dropout (Gal & Ghahramani, 2015) in the recurrent layers, with a dropout probability of 0.5. Optimization is performed using the Adam optimizer (Kingma & Ba, 2014), with a fixed learning rate of 1e-3.

## 3.1 DATA

We use electronic health records, with IRB approval, from the STARR Data Warehouse (previously known as STRIDE) for training and evaluation (Lowe et al., 2009). The Warehouse contains de-identified data for over 3 million patients (about 2.6% having a recorded date of death), spanning approximately 27 years. Each timestep in the sequence for a patient corresponds to all the data in the EHR for a given day. Only days having any data have a corresponding timestep in the sequence for each patient. We use diagnostic codes, medication order codes, lab test order codes, encounter type codes, and demographics (age and gender). Each code has a randomly initialized embedding vector as a trainable parameter. The set of 3 million patients, correspond to 51 million overall timesteps, and was randomly split in the ratio 8:1:1 into train, validation and test splits.

## 3.2 RESULTS

We first verify that all models are reasonably well-calibrated (Figure 4). Both the coefficient of variation and the Survival-AUPRC metrics suggest that the Survival-CRPS with interval censoring yields the sharpest prediction distributions (Table 1). Inspecting the mass past 120 years of age shows

that a naively trained prediction model with maximum likelihood can assign more than 75% of the mass to unreasonable regions, which is highly undesirable for the purpose of prediction. We note that this behavior is largely due to low prevalence of uncensored examples, which is typical in real world EHR data sets. As a result, the loss for the censored examples, which can be minimized by pushing mass as far away to the right as possible, dominates the small number of uncensored examples.

By predicting an entire distribution over time to death, the same model can be used to make classification predictions at various time points, highlighting the flexibility of our approach. When evaluated at 6 month, 1 year, and 5 year probabilistic predictions of mortality, our model remains well-calibrated with high discriminative ability (Appendix G, Figure 6).

In the interest of reproducibility, we run similar experiments on the publically available MIMIC-III dataset (Johnson et al., 2016) (Appendix H), and have published our corresponding source code [1].

## 4 RELATED WORK

Recent works have demonstrated potential to significantly improve patient care by making predictions with deep learning models on EHR data (Avati et al., 2017; Rajkomar et al., 2018), but these have been limited in treating the task as binary classification over a fixed time frame. Predicting survival curves instead of dichotomous outcomes has been explored (Yu et al., 2011; Lee et al., 2018), but only over finite length horizons. Deep survival analysis (Ranganath et al., 2016) has been proposed, but is limited to a fixed shape Weibull (bypassing the concerns we raised about stability, but limited in expressivity). The work by Yang et al. (2017) is similar to ours in terms of using an RNN and log-normal noise distribution, but limited to MLE training. DeepSurv (Katzman et al., 2018) uses a Cox proportional hazards model, which similarly makes a set of inflexible assumptions. The WTTE-RNN (Martinsson, 2016) model has a similar network architecture to ours, but is also limited to a Weibull distribution. All aforementioned models have only been optimized for maximum likelihood, instead of more robust proper scoring rules. The CRPS scoring rule has been used with Neural Networks in (Rasp & Lerch, 2018). Work in (Alaa & van der Schaar, 2017) also predicts full survival curves specific to a patient, but the use of GPs makes it difficult to scale to millions of patients. The work in Miscouridou et al. (2018) predict survival curves (both non-parametric, and flexible flow based parametric curves) while also handling missing covariates. Another recent work Chapfuwa et al. (2018) uses adversarial training for survival prediction. It has been shown that modern neural networks can be uncalibrated, and the work by Guo et al. (2017) suggests ways to improve calibration (though the work focuses on classification).

## 5 CONCLUSION

Better survival prediction models can be built by exploring objectives beyond maximum likelihood and evaluation metrics that assess the holistic quality of predicted distributions, instead of point estimates. We introduce the Survival-CRPS objective, motivated by the fact that the CRPS scoring rule is known to yield sharp prediction distributions while maintaining calibration. There are perhaps others scoring rules that work better, leaving avenues for future work. To evaluate, we introduce the Survival-AUPRC metric, which captures the degree to which a prediction distribution concentrates around the observed time of event. We demonstrate success in large-scale survival prediction by using a deep recurrent model employing a log-normal parameterization. By predicting an entire distribution for time-to-event, we circumvent issues associated with binary classification. Meanwhile, our model still yields accurate predictions when evaluated as dichotomous outcomes at particular times. The impact of having meaningfully accurate survival models is tremendous, especially in healthcare. We hope our work will be useful to those looking to build and deploy such models.

ACKNOWLEDGMENTS

We thank the PyTorch team, particularly for the `erf` implementation that allowed use of the log-normal distribution. We thank Baran Sandor, Sebastian Lerch, Alejandro Schuler, Jeremy Irvin, and Russell Greiner for valuable feedback.

---

[1] http://github.com/anonymoususer/anonymous.git

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

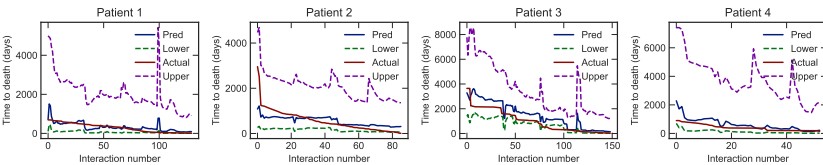

Figure 5: Median predicted time to death (with 95% intervals) for individual patients from the interval-censored Survival-CRPS model. Our model gives more confident predictions upon repeated interactions between patients and the EHR. True times to death generally lie within predicted intervals.

Tilmann Gneiting and Matthias Katzfuss. Probabilistic Forecasting. *Annual Review of Statistics and Its Application*, 1(1):125–151, 2014. doi: 10.1146/annurev-statistics-062713-085831.

Tilmann Gneiting and Adrian E. Raftery. Strictly proper scoring rules, prediction, and estimation. *Journal of the American Statistical Association*, 102(477):359–378, 2007. ISSN 01621459.

Tilmann Gneiting and Roopesh Ranjan. Comparing density forecasts using threshold- and quantile-weighted scoring rules. *Journal of Business & Economic Statistics*, 29(3):411–422, 2011. doi: 10.1198/jbes.2010.08110.

Tilmann Gneiting, Fadoua Balabdaoui, and Adrian E. Raftery. Probabilistic forecasts, calibration and sharpness. *Journal of the Royal Statistical Society. Series B: Statistical Methodology*, 69(2): 243–268, 4 2007. ISSN 1369-7412. doi: 10.1111/j.1467-9868.2007.00587.x.

Tilmann Gneiting, Larissa I. Stanberry, Eric P. Grimit, Leonhard Held, and Nicholas A. Johnson. Assessing probabilistic forecasts of multivariate quantities, with an application to ensemble predictions of surface winds. *TEST*, 17(2):211, Jul 2008. ISSN 1863-8260. doi: 10.1007/s11749-008-0114-x. URL https://doi.org/10.1007/s11749-008-0114-x.

David C. Goff, Donald M. Lloyd-Jones, Glen Bennett, Sean Coady, Ralph B. D'Agostino, Raymond Gibbons, Philip Greenland, Daniel T. Lackland, Daniel Levy, Christopher J. O'Donnell, Jennifer G. Robinson, J. Sanford Schwartz, Susan T. Shero, Sidney C. Smith, Paul Sorlie, Neil J. Stone, and Peter W. F. Wilson. 2013 ACC/AHA Guideline on the Assessment of Cardiovascular Risk: A Report of the American College of Cardiology/American Heart Association Task Force on Practice Guidelines. *Circulation*, 129(25 suppl 2):S49–S73, June 2014. ISSN 0009-7322, 1524-4539. doi: 10.1161/01.cir.0000437741.48606.98.

Jon Ketil Grønnesby and Ørnulf Borgan. A method for checking regression models in survival analysis based on the risk score. *Lifetime Data Analysis*, 2(4):315–328, Dec 1996. ISSN 1572-9249. doi: 10.1007/BF00127305. URL https://doi.org/10.1007/BF00127305.

Chuan Guo, Geoff Pleiss, Yu Sun, and Kilian Q. Weinberger. On calibration of modern neural networks. *CoRR*, abs/1706.04599, 2017. URL http://arxiv.org/abs/1706.04599.

Frank E. Harrell, Jr. *Regression Modeling Strategies*. Springer-Verlag, Berlin, Heidelberg, 2006. ISBN 0387952322.

David W. Hosmer, Stanley Lemeshow, and Susanne May. *Applied Survival Analysis: Regression Modeling of Time to Event Data: Second Edition*. Wiley Blackwell, 10 2011. ISBN 9780470258019. doi: 10.1002/9780470258019.

Wei L. J. The accelerated failure time model: A useful alternative to the cox regression model in survival analysis. *Statistics in Medicine*, 11(14-15):1871–1879. doi: 10.1002/sim.4780111409.

Alistair E. W. Johnson, Tom J. Pollard, Lu Shen, Li-wei H. Lehman, Mengling Feng, Mohammad Ghassemi, Benjamin Moody, Peter Szolovits, Leo A. Celi, and Roger G. Mark. MIMIC-III, a freely accessible critical care database. *Scientific Data*, 3:160035+, May 2016. ISSN 2052-4463. doi: 10.1038/sdata.2016.35.

Jared Katzman, Uri Shaham, Jonathan Bates, Alexander Cloninger, Tingting Jiang, and Yuval Kluger. DeepSurv: Personalized Treatment Recommender System Using A Cox Proportional Hazards Deep Neural Network. *BMC Medical Research Methodology*, 18(1), December 2018. ISSN 1471-2288. doi: 10.1186/s12874-018-0482-1. arXiv: 1606.00931.

Diederik P. Kingma and Jimmy Ba. Adam: A method for stochastic optimization. *CoRR*, abs/1412.6980, 2014. URL http://arxiv.org/abs/1412.6980.

Francis Lau, G. Michael Downing, Mary Lesperance, Jack Shaw, and Craig Kuziemsky. Use of palliative performance scale in end-of-life prognostication. *Journal of Palliative Medicine*, 9(5): 1066–1075, 10 2006. ISSN 1096-6218. doi: 10.1089/jpm.2006.9.1066.

Changhee Lee, William Zame, and Jinsung Yoon. DeepHit: A Deep Learning Approach to Survival Analysis with Competing Risks. *AAAI*, pp. 8, 2018.

Henry J Lowe, Todd A Ferris, Penni M Hernandez Nd, and Susan C Weber. STRIDE – An Integrated Standards-Based Translational Research Informatics Platform. *AMIA Annual Symposium Proceedings*, pp. 391–395, 2009.

Egil Martinsson. A model for sequential prediction of time-to-event in the case of discrete or continuous censored data, recurrent events or time-varying covariates. pp. 103, 2016.

Xenia Miscouridou, Adler J. Perotte, Noémie Elhadad, and Rajesh Ranganath. Deep survival analysis : Nonparametrics and missingness. 2018.

Seyedeh Atefeh Mohammadi, Morteza Rahmani, and Majid Azadi. Optimization of continuous ranked probability score using PSO, 2015.

Seyedeh Atefeh Mohammadi, Morteza Rahmani, and Majid Azadi. Meta-heuristic CRPS minimization for the calibration of short-range probabilistic forecasts. *Meteorology and Atmospheric Physics; Wien*, 128(4):429–440, August 2016. ISSN 01777971. doi: http://dx.doi.org/10.1007/s00703-015-0426-9.

Adam Paszke, Sam Gross, Soumith Chintala, Gregory Chanan, Edward Yang, Zachary DeVito, Zeming Lin, Alban Desmaison, Luca Antiga, and Adam Lerer. Automatic differentiation in pytorch. 2017.

Cox D. R. Regression models and life tables. *Journal of the Royal Statistic Society*, B(34):187–202, 1972.

Alvin Rajkomar, Eyal Oren, Kai Chen, Andrew M. Dai, Nissan Hajaj, Peter J. Liu, Xiaobing Liu, Mimi Sun, Patrik Sundberg, Hector Yee, Kun Zhang, Gavin E. Duggan, Gerardo Flores, Michaela Hardt, Jamie Irvine, Quoc Le, Kurt Litsch, Jake Marcus, Alexander Mossin, Justin Tansuwan, De Wang, James Wexler, Jimbo Wilson, Dana Ludwig, Samuel L. Volchenboum, Katherine Chou, Michael Pearson, Srinivasan Madabushi, Nigam H. Shah, Atul J. Butte, Michael Howell, Claire Cui, Greg Corrado, and Jeff Dean. Scalable and accurate deep learning for electronic health records. *arXiv:1801.07860 [cs]*, January 2018. arXiv: 1801.07860.

Prajit Ramachandran, Barret Zoph, and Quoc V. Le. Searching for activation functions. *CoRR*, abs/1710.05941, 2017. URL http://arxiv.org/abs/1710.05941.

Rajesh Ranganath, Adler Perotte, Noémie Elhadad, and David Blei. Deep Survival Analysis. *arXiv:1608.02158 [cs, stat]*, August 2016. arXiv: 1608.02158.

Stephan Rasp and Sebastian Lerch. Neural networks for post-processing ensemble weather forecasts. abs/1805.09091, 2018. URL https://arxiv.org/abs/1805.09091.

Leonard J. Savage. Elicitation of personal probabilities and expectations. *Journal of the American Statistical Association*, 66(336):783–801, 1971. ISSN 01621459.

Eli Sherman, Hitinder S. Gurm, Ulysses J. Balis, Scott R. Owens, and Jenna Wiens. Leveraging Clinical Time-Series Data for Prediction: A Cautionary Tale. In *AMIA 2017, American Medical Informatics Association Annual Symposium, Washington, DC, November 4-8, 2017*, 2017.

Nitish Srivastava, Geoffrey Hinton, Alex Krizhevsky, Ilya Sutskever, and Ruslan Salakhutdinov. Dropout: A simple way to prevent neural networks from overfitting. *J. Mach. Learn. Res.*, 15(1): 1929–1958, January 2014. ISSN 1532-4435.

Hajime Uno, Tianxi Cai, Lu Tian, and L. J. Wei. Evaluating Prediction Rules for t-Year Survivors with Censored Regression Models. *Journal of the American Statistical Association*, 102(478): 527–537, 2007. ISSN 0162-1459.

Yinchong Yang, Peter A. Fasching, and Volker Tresp. Modeling progression free survival in breast cancer with tensorized recurrent neural networks and accelerated failure time models. In Finale Doshi-Velez, Jim Fackler, David Kale, Rajesh Ranganath, Byron Wallace, and Jenna Wiens (eds.), *Proceedings of the 2nd Machine Learning for Healthcare Conference*, volume 68 of *Proceedings of Machine Learning Research*, pp. 164–176, Boston, Massachusetts, 18–19 Aug 2017. PMLR. URL http://proceedings.mlr.press/v68/yang17a.html.

Chun-Nam Yu, Russell Greiner, Hsiu-Chin Lin, and Vickie Baracos. Learning Patient-Specific Cancer Survival Distributions as a Sequence of Dependent Regressors. In J. Shawe-Taylor, R. S. Zemel, P. L. Bartlett, F. Pereira, and K. Q. Weinberger (eds.), *Advances in Neural Information Processing Systems 24*, pp. 1845–1853. Curran Associates, Inc., 2011.

APPENDIX

A. INTEGRAL IDENTITIES

Let $\Phi_{\mu,\sigma^2}(z)$ be the CDF of a Gaussian distribution with mean $\mu$ and variance $\sigma^2$. Hence $\Phi_{\mu,\sigma^2}(\log z)$ is the CDF of a log-normal distribution with mean $\mu$ and variance $\sigma^2$. For some integer $K$ (typically 32 in our experiments), we define $I$ to be the following integral, approximated by the trapezoidal rule:

$$I_{\mu,\sigma^2}(y,g) = \int_0^y \Phi_{\mu,\sigma^2}(\log z)^2 g(z)dz$$

$$\approx \sum_{k=0}^{K-1} \frac{1}{2}\left[\Phi_{\mu,\sigma^2}(\log z_{k+1})^2 g(z_{k+1}) + \Phi_{\mu,\sigma^2}(\log z_k)^2 g(z_k)\right](z_{k+1} - z_k)$$

where $0 = z_0 < z_1 < ... < z_K = y$ and $g$ is a function. We further define

$$I^+_{\mu,\sigma^2}(y) = I_{\mu,\sigma^2}(y, z \mapsto z),$$
$$I^-_{\mu,\sigma^2}(y) = I_{-\mu,\sigma^2}(1/y, z \mapsto 1/z^2).$$

B. SURVIVAL-CRPS FOR LOG-NORMAL (RIGHT-CENSORED)

For a general continuous prediction distribution $F$, with actual time to outcome $y \in \mathbb{R}_+$, and censoring indicator $c$, we generalize the CRPS to the Right Censored Survival CRPS score as:

$$\mathcal{S}_{\text{CRPS-RIGHT}}(F, (y, c)) = \int_{-\infty}^{\infty} (F(z)\mathbb{1}\{z \leq \log y \cup c = 0\} - \mathbb{1}\{z \geq \log y \cap c = 0\})^2 dz$$

$$= \int_{-\infty}^{\tilde{y}} F(z)^2 dz + (1-c)\int_{\tilde{y}}^{\infty}(F(z) - 1)^2 dz.$$

In the above expression $F$ would generally be in the family of continuous distributions over the entire real line (eg. Gaussian). Alternately, one could also use a family of distributions over the positive reals (e.g log-normal), in which case the Survival CRPS becomes:

$$\mathcal{S}_{\text{CRPS-RIGHT}}(F, (y, c)) = \int_0^{\infty} (F(z)\mathbb{1}\{z \leq y \cup c = 0\} - \mathbb{1}\{z \geq y \cap c = 0\})^2 dz$$

$$= \int_0^y F(z)^2 dz + (1-c)\int_y^{\infty}(F(z) - 1)^2 dz.$$

For the case of $F$ being log-normal, the expression becomes

$$\mathcal{S}_{\text{CRPS-RIGHT}}(F_{\text{LN}(\mu,\sigma^2)}, (y, c)) = \int_0^y \Phi_{\mu,\sigma^2}(\log z)^2 dz + (1-c)\int_y^{\infty}(1 - \Phi_{\mu,\sigma^2}(\log z))^2 dz$$

$$= \int_0^y \Phi_{\mu,\sigma^2}(\log z)^2 dz + (1-c)\int_y^{\infty} \Phi_{-\mu,\sigma^2}(-\log z)^2 dz$$

$$= \int_0^y \Phi_{\mu,\sigma^2}(\log z)^2 dz + (1-c)\int_0^{1/y} \Phi_{-\mu,\sigma^2}(\log z)^2(1/z)^2 dz$$

$$= I^+_{\mu,\sigma^2}(y) + (1-c)I^-_{\mu,\sigma^2}(y).$$

C. SURVIVAL-CRPS FOR LOG-NORMAL (INTERVAL-CENSORED)

We further extend the Right Censored Survival CRPS to the case of interval censoring. This is particularly useful for all-cause mortality prediction where we assume a particular event must occur by time $\mathcal{T}$. Using the same notations as before, the Interval Censored Survival CRPS is:

$$\mathcal{S}_{\text{CRPS-INTVL}}(F, (y, c, \mathcal{T})) = \int_0^{\infty} (F(z)\mathbb{1}\{\{z \leq y \cup c = 0\} \cup z \geq \mathcal{T}\} - \mathbb{1}\{\{z \geq y \cap c = 0\} \cup z \geq \mathcal{T}\})^2 dz$$

$$= \int_0^y F(z)^2 dz + (1-c)\int_y^{\mathcal{T}}(F(z) - 1)^2 dz + \int_{\mathcal{T}}^{\infty}(F(z) - 1)^2 dz.$$

For the case of $F$ being log-normal, the expression becomes

$$
\mathcal{S}_{\text{CRPS-INTVL}}(F_{\text{LN}(\mu,\sigma^2)},(y,c,\mathcal{T})) = \int_0^y \Phi_{\mu,\sigma^2}(\log z)^2 dz + (1-c)\int_y^{\mathcal{T}}(1-\Phi_{\mu,\sigma^2}(\log z))^2 dz
$$

$$
+ \int_{\mathcal{T}}^\infty (1-\Phi_{\mu,\sigma^2}(\log z))^2 dz
$$

$$
= \int_0^y \Phi_{\mu,\sigma^2}(\log z)^2 dz + (1-c)\int_{1/\mathcal{T}}^{1/y}\Phi_{-\mu,\sigma^2}(\log z)^2 (1/z)^2 dz
$$

$$
+ \int_0^{1/\mathcal{T}}\Phi_{-\mu,\sigma^2}(\log z)^2 (1/z)^2 dz
$$

$$
= I_{\mu,\sigma^2}^+(y) + I_{\mu,\sigma^2}^-(\mathcal{T}) + (1-c)\left[I_{\mu,\sigma^2}^-(y) - I_{\mu,\sigma^2}^-(\mathcal{T})\right].
$$

## D. SURVIVAL-AUPRC FOR LOG-NORMAL (INTERVAL-CENSORED)

We start with the most general case (interval censoring). For a general continuous prediction distribution $F$ with an interval outcome $[L,U]$, we define the Survival-AUPRC as

$$
\text{Survival-AUPRC}(F,L,U) = \int_0^1 \left[F(U/t) - F(Lt)\right] dt.
$$

Specifically for the case of log-normal, where $\phi$ and $\Phi$ are PDF and CDF of $\mathcal{N}(0,1)$ respectively, and $\tilde{L} = \log L$ and $\tilde{U} = \log U$:

$$
\text{Survival-AUPRC}(F_{\text{LN}(\mu,\sigma^2)},L,U) = \int_0^1 \left[F_{\text{LN}(\mu,\sigma^2)}(U/t) - F_{\text{LN}(\mu,\sigma^2)}(Lt)\right] dt
$$

$$
= \int_0^1 \left[F_{\mathcal{N}(\mu,\sigma^2)}(\tilde{U} - \log t) - F_{\mathcal{N}(\mu,\sigma^2)}(\tilde{L} + \log t)\right] dt
$$

$$
(\text{substituting } s = \log t) = \int_{-\infty}^0 \left[F_{\mathcal{N}(\mu,\sigma^2)}(\tilde{U} - s) - F_{\mathcal{N}(\mu,\sigma^2)}(\tilde{L} + s)\right] e^s ds
$$

$$
= \left[F_{\mathcal{N}(\mu,\sigma^2)}(\tilde{U} - s) - F_{\mathcal{N}(\mu,\sigma^2)}(\tilde{L} + s)\right] e^s \Big|_{s=-\infty}^{s=0}
$$

$$
- \int_{-\infty}^0 \left[-f_{\mathcal{N}(\mu,\sigma^2)}(\tilde{U} - s) - f_{\mathcal{N}(\mu,\sigma^2)}(\tilde{L} + s)\right] e^s ds
$$

$$
= \left(F_{\mathcal{N}(\mu,\sigma^2)}(\tilde{U}) - F_{\mathcal{N}(\mu,\sigma^2)}(\tilde{L})\right)
$$

$$
+ \int_{-\infty}^0 \left[f_{\mathcal{N}(\mu,\sigma^2)}(\tilde{U} - s) + f_{\mathcal{N}(\mu,\sigma^2)}(\tilde{L} + s)\right] e^s ds
$$

$$
= \left(F_{\mathcal{N}(\mu,\sigma^2)}(\tilde{U}) - F_{\mathcal{N}(\mu,\sigma^2)}(\tilde{L})\right)
$$

$$
+ \int_{-\infty}^0 f_{\mathcal{N}(\mu,\sigma^2)}(\tilde{U} - s) e^s ds + \int_{-\infty}^0 f_{\mathcal{N}(\mu,\sigma^2)}(\tilde{L} + s) e^s ds
$$

$$
= \left(F_{\mathcal{N}(\mu,\sigma^2)}(\tilde{U}) - F_{\mathcal{N}(\mu,\sigma^2)}(\tilde{L})\right)
$$

$$
+ \int_{-\infty}^0 \frac{1}{\sigma}\phi\left(\frac{\tilde{U} - s - \mu}{\sigma}\right) e^s ds + \int_{-\infty}^0 \frac{1}{\sigma}\phi\left(\frac{\tilde{L} + s - \mu}{\sigma}\right) e^s ds
$$

$$
\left(\text{substituting } u = \frac{\tilde{U} - s - \mu}{\sigma}\right) = \left(F_{\mathcal{N}(\mu,\sigma^2)}(\tilde{U}) - F_{\mathcal{N}(\mu,\sigma^2)}(\tilde{L})\right)
$$

$$
+ \int_\infty^{\frac{\tilde{U}-\mu}{\sigma}} \frac{1}{\sigma}\phi(u) e^{\tilde{U}-\sigma u-\mu}(-\sigma)du + \int_{-\infty}^0 \frac{1}{\sigma}\phi\left(\frac{\tilde{L} + s - \mu}{\sigma}\right) e^s ds
$$

$$\left(\text{substituting } v = \frac{\tilde{L} + s - \mu}{\sigma}\right) = \left(F_{\mathcal{N}(\mu,\sigma^2)}(\tilde{U}) - F_{\mathcal{N}(\mu,\sigma^2)}(\tilde{L})\right)$$

$$+ \int_{\infty}^{\frac{\tilde{U}-\mu}{\sigma}} \frac{1}{\sigma} \phi(u) e^{\tilde{U}-\sigma u-\mu}(-\sigma)du + \int_{-\infty}^{\frac{\tilde{L}-\mu}{\sigma}} \frac{1}{\sigma} \phi(v) e^{v\sigma-\tilde{L}+\mu}\sigma dv$$

$$= \left(F_{\mathcal{N}(\mu,\sigma^2)}(\tilde{U}) - F_{\mathcal{N}(\mu,\sigma^2)}(\tilde{L})\right)$$

$$- e^{\tilde{U}-\mu} \int_{\infty}^{\frac{\tilde{U}-\mu}{\sigma}} \phi(u) e^{-\sigma u}du + e^{-\tilde{L}+\mu} \int_{-\infty}^{\frac{\tilde{L}-\mu}{\sigma}} \phi(v) e^{v\sigma}dv$$

$$\left(\text{using } \int e^{cx}\phi(x)dx = e^{\frac{c^2}{2}}\Phi(x-c)\right) = \left(F_{\mathcal{N}(\mu,\sigma^2)}(\tilde{U}) - F_{\mathcal{N}(\mu,\sigma^2)}(\tilde{L})\right)$$

$$+ \frac{U}{e^\mu}\left[e^{\frac{\sigma^2}{2}}\Phi(u+\sigma)\right]_{u=\infty}^{u=\frac{\tilde{U}-\mu}{\sigma}} + \frac{e^\mu}{L}\left[e^{\frac{\sigma^2}{2}}\Phi(v-\sigma)\right]_{v=-\infty}^{v=\frac{\tilde{L}-\mu}{\sigma}}$$

$$= \left(F_{\mathcal{N}(\mu,\sigma^2)}(\tilde{U}) - F_{\mathcal{N}(\mu,\sigma^2)}(\tilde{L})\right)$$

$$+ \frac{U}{e^\mu}\left[e^{\frac{\sigma^2}{2}}\Phi\left(\frac{\tilde{U}-\mu}{\sigma}+\sigma\right) - e^{\frac{\sigma^2}{2}}\right] + \frac{e^\mu}{L}\left[e^{\frac{\sigma^2}{2}}\Phi\left(\frac{\tilde{L}-\mu}{\sigma}-\sigma\right)\right]$$

$$= \left(F_{\mathcal{N}(\mu,\sigma^2)}(\tilde{U}) - F_{\mathcal{N}(\mu,\sigma^2)}(\tilde{L})\right)$$

$$+ e^{\frac{\sigma^2}{2}}\left[\frac{e^\mu}{L}\Phi\left(\frac{\tilde{L}-\mu}{\sigma}-\sigma\right) + \frac{U}{e^\mu}\left(1 - \Phi\left(\frac{\tilde{U}-\mu}{\sigma}+\sigma\right)\right)\right]$$

$$= \Phi_{\mu,\sigma^2}(\log U) - \Phi_{\mu,\sigma^2}(\log L)$$

$$+ e^{\frac{\sigma^2}{2}}\left[\frac{e^\mu}{L}\Phi\left(\frac{\log L-\mu}{\sigma}-\sigma\right) + \frac{U}{e^\mu}\Phi\left(-\frac{\log U-\mu}{\sigma}-\sigma\right)\right].$$

## E. SURVIVAL-AUPRC FOR LOG-NORMAL (RIGHT-CENSORED)

For a general continuous prediction distribution $F$ with an interval outcome $[L, \infty)$, we define Survival-AUPRC as

$$\text{Survival-AUPRC}(F, L) = \int_0^1 [1 - F(Lt)]\,dt.$$

Specifically for the case of log-normal, where $\Phi$ is the CDF of $\mathcal{N}(0,1)$, and $\tilde{L} = \log L$ (following Appendix-D),

$$\text{Survival-AUPRC}(F_{\text{LN}(\mu,\sigma^2)}, L) = \int_0^1 \left[1 - F_{\text{LN}(\mu,\sigma^2)}(Lt)\right]dt = 1 - \Phi_{\mu,\sigma^2}(\tilde{L}) + \frac{e^{\mu+\frac{\sigma^2}{2}}}{L}\Phi\left(\frac{\tilde{L}-\mu}{\sigma}-\sigma\right).$$

## F. SURVIVAL-AUPRC FOR LOG-NORMAL (UNCENSORED)

For a general continuous prediction distribution $F$ with a point outcome $y$, we define Survival-AUPRC

$$\text{Survival-AUPRC}(F, y) = \int_0^1 [F(y/t) - F(yt)]\,dt.$$

Specifically for the case of log-normal, where $\Phi$ is the CDF of $\mathcal{N}(0,1)$, and $\tilde{y} = \log y$ (following Appendix-D),

$$\text{Survival-AUPRC}(F_{\text{LN}(\mu,\sigma^2)}, y) = \int_0^1 \left[F_{\text{LN}(\mu,\sigma^2)}(y/t) - F_{\text{LN}(\mu,\sigma^2)}(yt)\right]dt$$

$$= e^{\frac{\sigma^2}{2}}\left[\frac{e^\mu}{y}\Phi\left(\frac{\tilde{y}-\mu}{\sigma}-\sigma\right) + \frac{y}{e^\mu}\Phi\left(-\frac{\tilde{y}-\mu}{\sigma}-\sigma\right)\right].$$

## G. EVALUATION AS BINARY OUTCOME

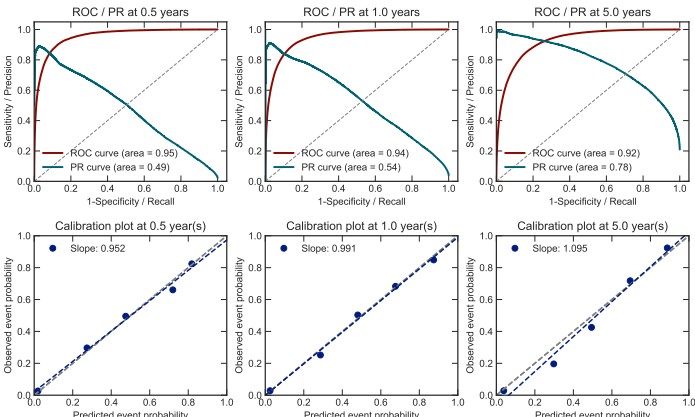

Figure 6: Discrimination and calibration of predictions from the interval-censored Survival-CRPS model, evaluated as predictions for a dichotomous outcome at 6 months, 1 year, and 5 years.

## H. EXPERIMENTS ON THE MIMIC-III DATASET

On the MIMIC-III dataset (Johnson et al., 2016), we built a feed forward neural network that takes in $51015$ hospital admissions in the dataset ($70.1\%$ censored) and makes predictions at the time of discharge. There is only one time of prediction per patient, so a recurrent model was not used. We removed admissions where the patient's age was obfuscated or where the patient's discharge time occurred after their recorded date of death. As features, we used demographics (age and gender) and embedded diagnostic codes into a 128-dimensional space.

Table 2: For MIMIC-III, metrics measuring sharpness and calibration for models trained on the right-censored and interval-censored variants of the maximum likelihood and Survival-CRPS objectives.

| Metric | MLE-RIGHT | MLE-INTVL | CRPS-RIGHT | CRPS-INTVL |
|---|---|---|---|---|
| Calibration slope | $1.190 \pm 5e\text{-}3$ | $0.932 \pm 9e\text{-}3$ | $1.190 \pm 5e\text{-}3$ | $0.938 \pm 7e\text{-}3$ |
| Mean coefficient of variation | $4.062 \pm 0.039$ | $1.763 \pm 0.006$ | $4.062 \pm 0.035$ | $\mathbf{1.647 \pm 0.012}$ |
| Mean prob of survival to age 120 yrs | $0.035 \pm 4e\text{-}4$ | $0.007 \pm 2e\text{-}4$ | $0.035 \pm 4e\text{-}4$ | $\mathbf{0.001 \pm 2e\text{-}6}$ |
| Dead: mean Surv-AUPRC (uncen) | $0.266 \pm 2e\text{-}3$ | $0.338 \pm 4e\text{-}3$ | $0.266 \pm 3e\text{-}3$ | $\mathbf{0.348 \pm 4e\text{-}3}$ |
| Alive: mean Surv-AUPRC (intvl-cen) | $0.993 \pm 2e\text{-}4$ | $0.999 \pm 6e\text{-}5$ | $0.993 \pm 2e\text{-}4$ | $\mathbf{1.000 \pm 1e\text{-}5}$ |

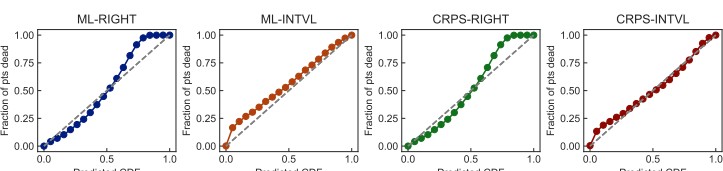

Figure 7: Calibration plots for each of the models in MIMIC-III. We compare predicted cumulative densities against observed event frequencies, evaluated at quantiles of predicted cumulative density. Right-censored observations are removed from consideration in quantiles past times of censoring, interval-censored observations are additionally re-introduced in quantiles corresponding to times past 120 years.

