# OpenReview forum: "Countdown Regression: Sharp and Calibrated Survival Predictions"
_ICLR.cc/2019/Conference_

### Official Review · AnonReviewer1 · 2018-11-02
**Mathematically elegant, insufficient experimentation**

**Rating:** 5
**Confidence:** 3

**Review:**

The paper proposes the use of Survival Continuous Ranked Probability score instead of maximum likelihood estimation for personalised probabilistic forecasts of time-to-event data, thus estimating a distribution over future time. The authors describe the evaluation their method using (1) proper scoring rule objectives; (2) evaluation of calibration using sharpness as a metric; (3) the survival precision recall curve. The authors then apply these techniques to predicting time-to-mortality using an RNN that takes EHR patient records to predict the probability of death at a given time point. It’s not clear how this is related to the Survival CRPS model or how this model is incorporated into the RNN.
Overall, this is an important framework for estimating personalised predictions of survival events for patients with interval-censored data. The authors present a well thought-out paper with clearly and realistically articulated modelling  assumptions. The authors also give an excellent critique of the underlying assumptions of current state-of-the-art survival methods. The authors are also to be commended for the mathematical elegance
Although the paper is very well written and extremely well structured, I struggled with the lack of experiments available in the paper.
The text embedded in Figure 3 is too small.
The results section is somewhat sparse. Although the mathematical formulation is well-motivated and structured, it’s not clear what the contribution of this work is. The difference between CRPS-INTVL and MLE-INTVL is incremental and it’s unclear what the significant benefits are of CRPS vs MLE. What would the interpretation of these differences in a real-world setting?

---

> ### Author Response · Authors · 2018-11-25
> **reply**
>
> Dear AnonReviewer1,
>
> Thank you for taking the time to review our work and offer constructive feedback. We hope to address some of your concerns in the following reply.
>
> You mention that it’s unclear how the RNN taking EHR patient records is related to the Survival CRPS scoring rule. To clarify, we use the same RNN architecture to compare different loss functions: the right-censored and interval-censored versions of the MLE and CRPS scoring rules. We believe our primary contribution in this work is not the RNN model architecture (which is similar to the WTTE-RNN paper), but a robust proper scoring rule for the survival prediction setting.
>
> We hope to clarify that the contribution of our work is a framework for building predictive time-to-event models on real-world EHR datasets. This includes: the Survival-CRPS as a scoring rule, the RNN model architecture, and the Survival-AUPRC metric for evaluation.
>
> You mention that the interpretation of differences between CRPS and MLE in the real-world setting is unclear. We hope to clarify that our set of experiments (both in the main text and Appendix-H) are built on real-world EHR datasets, and it is the presence of heavy censoring in these datasets that motivates Survival-CRPS scoring rule as an alternative to MLE. We observe that our experimental results show that the MLE-INTVL and CRPS-INTVL scoring rules yield statistically significant differences in the sharpness of predicted distributions, particularly as measured by coefficient of variation and Survival AUPRC (Table 1).
>
> We hope that you would be kind enough to revisit your evaluation based on our reply. Thank you again for your time spent reviewing and constructive feedback on our work.

---

### Official Review · AnonReviewer3 · 2018-11-05
**COUNTDOWN REGRESSION: SHARP AND CALIBRATED SURVIVAL PREDICTIONS**

**Rating:** 4
**Confidence:** 4

**Review:**

The authors introduce an extension of Continuous Ranked Probability Scores (CRPS) to the time-to-event setting termed Survival-CRPS for both right censored and interval-censored event data. Further, the authors introduce a scale agnostic Survival-AUPRC evaluation metric that is analogous to the precision-recall curve used in classification and information retrieval systems/models.

The claim that that the proposed approach constitutes the first time a scoring rule other than maximum likelihood seems too strong, unnecessary and irrelevant to the value of the presented work.

It is not clear how did the authors handle the irregularity (in time) of EHR encounters in the context of an RNN specification. Also, if the RNN specification considered is similar to Martinsson, 2016, why this wasn't considered as a competing model in the experiments?

In Table 1 , it is not clear what the error bars are also they seem too small.

The proposed approach addresses important questions in time-to-event modeling, namely, calibration and interval censoring. Although the connection with CRPS is interesting (first of the two equations in page 3), it is quite similar to an accelerated failure time formulation, which for a log-normal specification is standard and popular due to similar reasons to those highlighted by the authors, but not mentioned in the related work. The interval censoring is also interesting, though straightforward and perhaps not as relevant in more general time-to-event settings where events other than age are considered.

The Survival-AUPRC is not sufficiently motivated. Without motivation or an intuition of why it should be used/preferred, it seems disconnected from the rest of the paper and its contributions.

Without a more comprehensive evaluation that includes additional datasets and competing models (described in the Related Work Section) it is difficult to assess the value of the proposed approach.

---

> ### Author Response · Authors · 2018-11-25
> **reply**
>
> Dear AnonReviewer3,
>
> We thank you for the time you spent reviewing and for the thoughtful comments. We hope to address some of your concerns in the following reply.
>
> You point out that our claim that the approach constitutes the first time a scoring rule other than maximum likelihood has been used seems too strong. We hope to clarify that we believed it is the first time a scoring rule other than maximum likelihood has been applied to the *survival prediction* setting. However we have recently come across works such as adversarial time to event models, and therefore will remove the claim from the paper. Thank you! [Update: changed the sentence]
>
> You also mention that it is unclear how irregularity (in time) of EHR encounters was handled by the RNN model. We hope to clarify that each recorded interaction between a patient and the EHR is considered a timestep in the sequence input to the RNN, regardless of how long has passed between each interaction (section 2.4). Moreover, the age of the patient at each timestep is treated as a feature for each timestep, which allows the RNN to account for passage of time. This approach naturally handles the irregularity of time between EHR encounters.
>
> We believe our primary contribution is not the model architecture, but a robust proper scoring rule for the survival prediction setting. We therefore do not compare our model against the WTTE-RNN architecture, but instead treat the RNN model as fixed and focus on a comparison between the CRPS and MLE scoring rules.
>
> You make a very valid point that in Table 1, it is unclear what the role is of the error bars. We clarify that these error bars are 95% confidence intervals for the corresponding metrics on the test set, constructed through bootstrap resampling. We will add this explanation to the main text. Our test set is quite large, and hence the intervals are pretty tight.
>
> You mention that the Survival-AUPRC is not sufficiently motivated. We will work on the phrasing, but its primary motivation remains: a holistic approach to evaluating the quality of a time to event prediction distribution. In contrast, measures such as relative absolute error or C-statistic only take into account predicted point estimates (such as predicted median time to event). And the other measures we report in our work (calibration slope, coefficient of variation) only measure calibration *or* sharpness, but are unable to capture both at once.
>
> With respect to further experimentation, we emphasize that we believe our contributions are most useful for training models on real world EHR datasets with heavy censoring. Our discussions with other hospitals have led us to believe that data from most EHRs have similar levels of censoring / label prevalence as our main result. This makes us confident that our work will be very relevant for those building models for production in the real world. Yet, unfortunately, most publicly available datasets are either too small, or have much lower levels of censoring thereby making MLE itself work well enough (though MLE’s predictive performance on datasets with real world levels of censoring is very bad). We would very much like to obtain results from a second real-world data set where the strengths of our work shows best, but it is a time consuming process to get access to hospital EHRs.
>
> We hope that you would be kind enough to revisit your evaluation based on our reply. Thank you again for your time spent reviewing and constructive feedback on our work.

---

### Official Review · AnonReviewer2 · 2018-11-13
**interesting topic, insufficient experiments**

**Rating:** 4
**Confidence:** 4

**Review:**

 My main concern is that the authors fail to compare their appproach to any of the modelling approaches discussed in the related works section. In particular, as mentioned by the authors the WTTTE-RNN has a similar architecture and thus would have been a crucial baseline for comparisons.
 Furthermore, I would have liked to see an evaluation on more datasets, especially since the data in Appendix H indicate that the proposed approach is only marginally better than MLE-based model fitting.
Finally, in addition to the metrics presented, conventional metrics such as the C-statistic would have been interesting.

I further miss a discussion of alternative approaches to achieve well calibrated scores, especially posthoc calibration using the validation set as discussed in Guo et al, ICML 2017.

Related work is incomplete, for example the use of tensor-trains in RNNs to model EHR data (Yang et al) - would the proposed approach not benefit for the use of such tensorization to better model the high-dimensional, sparse EHR data?


references:
Guao et al, On Calibration of Modern Neural Networks, ICML 2017
Yang et al, Modeling progression free survival in breast cancer with tensorized recurrent neural networks and accelerated failure time models, Machine Learning for Healthcare Conference 2017

---

> ### Author Response · Authors · 2018-11-25
> **reply**
>
> Dear AnonReviewer2,
>
> Thank you for taking time to review our paper and offering constructive feedback. We truly appreciate your time and effort towards this. We hope to address some of your questions and concerns in our following message.
>
> As you describe, your main concern is the lack of comparison with WTTE-RNN since the models are similar. In our opinion, the similarity of the model is incidental. We believe our primary contribution is not the model architecture, but a robust proper scoring rule for the survival prediction setting. This way, we believe the “competitor” is MLE, and fair comparisons are best done in the form of holding any model fixed, and replacing MLE with Survival-CRPS. In this view, we think of WTTE-RNN as yet another model, currently trained by MLE, which could alternatively be trained via Survival-CRPS. We did in fact try this experiment, but for reasons described in the section “2.5 Choice of Noise Distribution”, fitting Weibull-like distributions with both shape and scale parameters jointly over massive noisy datasets can be extremely challenging (either via MLE or CRPS).
>
> Your second point about evaluation on more datasets is very valid. The results in Appendix H are not as pronounced as the results in the main paper - this is primarily because of the rate of censoring. Our method shows greatest contrast over MLE when the rate of censoring is very high (e.g. 97.4% in our main results, whereas only 70% in the Appendix H result). Most real world data sets from EHRs (based on conversations with at least 3 other hospitals) have similar level of censoring / label prevalence as our main result. This makes us pretty confident that our work will be very relevant for those building models for production in the real world. Yet, unfortunately, most publicly available datasets are either too small, or have much lower levels of censoring thereby making MLE itself work well enough (though MLE’s predictive performance on real world level censoring is very bad). We would very much like to obtain results from a second real-world data set where the strengths of our work shows best, but it is a time consuming process to get access to hospital EHRs.
>
>
> Another point you mention is reporting more traditional metrics such as C-statistic. We intentionally left out reporting C-statistic, as a way to force our readers to think beyond it. While it is true that C-statistic is commonly reported, we feel it is a very poor representation of the clinical utility of a model. For the purposes of clinical decision making, our opinion is that calibration is a far more important and relevant metric.
>
> That being said, if you still feel strongly about it, we will be happy to include C-statistic results. The C-statistic numbers across the 4 categories look very similar.
>
> You also mention about Guo et. al (2017) work summarizing post-hoc techniques to improve calibration. We have a couple of thoughts on that: First, their work addresses a slightly different concern - about re-calibrating uncalibrated predictions (i.e, even when using a proper scoring rule, generalization of uncertainty estimates may be poor). In principle, their work is somewhat complementary to our goals (which is, striving to improving sharpness while ensuring calibration -- its generalization aspect is somewhat a distinct problem). Secondly, the methods they describe are focused on binary classification with fully observed labels. These methods do not quite readily apply to real valued prediction with different kinds of censoring.
>
> Yet, we agree that we should address that work in our paper. Thank you for the suggestion. [Update: the work is now cited appropriately]
>
> You make good suggestions about other related work. We will certainly go through that paper and appropriately cite it as related work. Thank you for the pointer. [Update: citations added]
>
> Finally, we thank you once again for your constructive feedback. We hope you will be generous enough to revisit your evaluation based on our replies.

---

### Official Review · AnonReviewer4 · 2018-11-16

**Rating:** 4
**Confidence:** 5

**Review:**

The paper presents a new loss function for survival analysis based
on proper scoring functions to less then penalty wrong predictions
that are confident make under the log-loss. The paper is interesting
however the benefit over the traditional maximum likelihood estimator is small and the writing needs a bunch of work. I would also like to see an eval on data with far less censoring.


A couple of comments

1) EHRs have only been generally adopted in the last couple of years. Only A couple of places have more.

2) Binary classifier citation on page 1 (Avati, Rajkomar) should also cite the plethora of recent machine
learning for healthcare results in this field

3) Likelihoods are calibrated (as is any error measured by a proper scoring loss)

4) There are other methods to fit survival functions such as "Adversarial Time-to-Event Modeling"
by Chapfuwa in ICML 2018. There are probably also moment methods

5) I think the evaluation might also want utilty because sharpness is a utility claim

6) Some of the statements in the writing are funny like probability distributions are uniquely identified by parameters. I'm not sure this is true with neural nets with symmetries. The paper doesn't need such claims

7) Instead of log-normals, I would like to see something nonparametric like the categoricals  used for maximum likelihood estimation without latents in the limiting model in "Deep Survival Analysis: Missingness and Nonparametrics" by Miscouridou at MLHC 2018

---

> ### Author Response · Authors · 2018-11-25
> **reply**
>
> Dear AnonReviewer4,
>
> Thank you very much for your insightful and thoughtful review. We agree with most of your comments. In particular:
>
> We need to include more references related binary classification of time to event, and we will.
> Thank you for bringing to our attention the “Adversarial Time-to-Event Modeling” paper. We were not aware of it, and looks very relevant to our work. At the very least we will make a quick pass to appropriately cite it as related work. [Update: we have.]
> The way we say probability distributions are uniquely identified by parameters is indeed incorrect. In that statement we were not referring to distributions parametrized by models, but in either case the paper is better off without the clause. [Update: we have.]
>
> We would like to share a few observations based on your comments.
>
> We do have experiments with much lower rate of censoring in the Appendix (H). Those set of experiments is run on publicly available data (MIMIC-III) and we have also shared the code and data pipeline in a public GitHub repo in the interest of reproducibility and evaluation. The data there is only 70% censored, and our loss function shows promising results in comparison to MLE based predictive performance. However we have not included the link in the paper for anonymity. We understand reviewers are not obliged to see the appendix, and we will consider it very generous of you if you could have a look at that section (hoping that it addresses your concern about experiments data with far less censoring).
> Thank you for highlighting utility measures. It would certainly be interesting to include evaluation on utility. We chose to limit our analysis purely to predictive performance and stay agnostic to interventions in this study.
> Non-parametric approaches are certainly interesting, and we have some work in-progress along this front. Survival prediction with Categorical distribution like the approach described in Miscouridou et. al. (2018), in our opinion, fit better with latent variable models, since smoothness can be naturally applied with priors (like the linear dynamical system approach in their work). We feel enforcing smoothness over categoricals with fully observed / conditional models such as ours can often be ad-hoc, and we are hoping to address this in our separate in-progress work. We feel including non-parametric approaches in our current work might be a little beyond its scope. [Update: work is now cited]
>
> We also hope you will be kind enough to revisit your evaluation based on our reply. Most importantly, we thank you again for your highly insightful feedback and new pointers.

---

### Meta-Review · Area_Chair1 · 2018-12-14
**Meta-Review for Countdown Regression paper**

**Confidence:** 5
**Recommendation:** Reject

**Metareview:**

All reviewers agree to reject. While there were many positive points to this work, reviewers believed that it was not yet ready for acceptance.